# Sirtuin1-Mediated Deacetylation of Hypothalamic TTF-1 Contributes to the Energy Deficiency Response

**DOI:** 10.3390/ijms241512530

**Published:** 2023-08-07

**Authors:** Dasol Kang, Hye Rim Yang, Dong Hee Kim, Kwang Kon Kim, Bora Jeong, Byong Seo Park, Jeong Woo Park, Jae Geun Kim, Byung Ju Lee

**Affiliations:** 1Department of Biological Sciences, College of Natural Sciences, University of Ulsan, Ulsan 44610, Republic of Korea; lainef7@ulsan.ac.kr (D.K.); gainlydh@ulsan.ac.kr (D.H.K.); fromfriend77@gmail.com (K.K.K.); boraring@naver.com (B.J.); jwpark@ulsan.ac.kr (J.W.P.); 2Division of Life Sciences, College of Life Sciences and Bioengineering, Incheon National University, Incheon 22012, Republic of Korea; hr.yang0414@gmail.com (H.R.Y.); 2021s135@inu.ac.kr (B.S.P.); 3Division of Gastroenterology and Hepatology, Department of Medicine, Stanford University, Stanford, CA 94305, USA

**Keywords:** TTF-1, sirtuin1, deacetylation, AgRP, POMC, energy deficiency

## Abstract

TTF-1 stimulates appetite by regulating the expression of agouti-related peptide (AgRP) and proopiomelanocortin (POMC) genes in the hypothalamus of starving animals. However, the mechanism underlying TTF-1’s response to decreased energy levels remains elusive. Here, we provide evidence that the NAD^+^-dependent deacetylase, sirtuin1 (Sirt1), activates TTF-1 in response to energy deficiency. Energy deficiency leads to a twofold increase in the expression of both Sirt1 and TTF-1, leading to the deacetylation of TTF-1 through the interaction between the two proteins. The activation of Sirt1, induced by energy deficiency or resveratrol treatment, leads to a significant increase in the deacetylation of TTF-1 and promotes its nuclear translocation. Conversely, the inhibition of Sirt1 prevents these Sirt1 effects. Notably, a point mutation in a lysine residue of TTF-1 significantly disrupts its deacetylation and thus nearly completely hinders its ability to regulate AgRP and POMC gene expression. These findings highlight the importance of energy-deficiency-induced deacetylation of TTF-1 in the control of AgRP and POMC gene expression.

## 1. Introduction

Thyroid transcription factor-1 (TTF-1) is a homeodomain-containing transcription factor expressed in the thyroid, lung, and brain [1,2]. In the rodent brain, TTF-1 is expressed in astrocytes [3,4] as well as certain neuronal cells in the hypothalamus, including those expressing gonadotropin-releasing hormone, proenkephalin, agouti-related peptide (AgRP), and proopiomelanocortin (POMC) [5,6,7]. TTF-1 binds to the 5′ flanking region of AgRP and POMC genes, exerting regulatory control over their gene expression [8]. Notably, during periods of low energy state in rodents, expression of TTF-1 in the hypothalamus is stimulated, leading to the upregulation of feeding behavior by influencing the expression of these specific target genes. Overnight fasting induces an increase in TTF-1 expression, subsequently stimulating the expression of orexigenic AgRP and inhibiting anorexigenic POMC genes. However, this fasting-induced modulation of AgRP and POMC gene expression is impeded when TTF-1 synthesis is inhibited by RNA interference [8,9]. These findings suggest that TTF-1 plays a crucial role in regulating appetite behavior by controlling the expression of target neuropeptides during conditions of low body energy [8]. However, it is largely unclear how TTF-1 expression responds to an energy deficiency.

The ratio of energy-dependent metabolites between adenosine triphosphate (ATP) and adenosine diphosphate (ADP), as well as between oxidized nicotinamide adenine dinucleotide (NAD^+^) and reduced NAD (NADH), undergoes dynamic changes in the hypothalamus in response to fluctuations in the body’s energy levels [10,11,12,13]. The intracellular ratio of these metabolites is closely correlated with energy-sensing mechanisms of intracellular regulators such as mammalian target of rapamycin and sirtuin1 (Sirt1) [14,15,16]. Sirt1 is a nutrient-sensitive NAD^+^-dependent class III deacetylase and plays an important role in energy balance [17,18].

Sirt1 expression in the hypothalamus is induced by diet restriction and a decrease in Sirt1 expression and activity results in a decreased food intake and body weight gain in lean rodents, whereas pharmacological activation of Sirt1 results in an increased food intake and body weight gain [19,20,21,22,23]. Sirt1 is essential in the AgRP and POMC neurons for metabolic functions [18,24]. Therefore, inhibition of Sirt1 specifically decreases energy intake and body weight by regulation of AgRP and POMC neurons [25,26,27]. Previous studies have shown that TTF-1 is an interaction partner of Sirt1 and Sirt1-induced deacetylation of TTF-1 results in a stimulation of TTF-1 activity for the regulation of orexin 2 receptor transcription [28]. However, it is unclear whether Sirt1-induced deacetylation is important in the response of TTF-1 to an energy deficiency, which results in an increased appetite through the regulation of AgRP and POMC gene expression in the hypothalamus.

In this study, we show that energy deficiency triggers the activation of hypothalamic NAD^+^–Sirt1 activity, leading to enhanced deacetylation of TTF-1. This deacetylation process is critical for the nuclear translocation and activity of TTF-1, which in turn regulates the expression of AgRP and POMC genes.

## 2. Results

### 2.1. Changes in Hypothalamic TTF-1 and Sirt1 Gene Expression Induced by Energy Deficiency

Given that hypothalamic TTF-1 and Sirt1 are recognized as crucial molecular components involved in the regulation of energy metabolism and their expression is influenced by the overall energy status of the body, we conducted an investigation to determine whether the gene expressions of both TTF-1 and Sirt1 are responsive to conditions of low energy availability (Figure 1). We observed that food deprivation for 24 h led to an increase in the expression of TTF-1 and Sirt1 in the mouse hypothalamus in vivo (Figure 1A–E). Similarly, in vitro experiments using cultured mHypoA 2/28 hypothalamic cells (denoted as mHypoA) in a serum-free starvation medium with low glucose (referred to as starvation medium) showed an elevation in TTF-1 and Sirt1 expression (Figure 1G–K). Additionally, both in vivo and in vitro conditions of energy deficiency resulted in an increased NAD^+^/NADH ratio (Figure 1F,L), consistent with previous reports on the brain and peripheral tissues under energy deprivation conditions [11,13,29]. Collectively, these findings indicate that the hypothalamic expression of TTF-1 and Sirt1, as well as the NAD^+^/NADH ratio, reflect changes in the body’s energy state.

### 2.2. Energy-Deficiency-Induced TTF-1 and Sirt1 Interaction and TTF-1 Deacetylation

Based on the evidence demonstrating the interaction between Sirt1 and TTF-1, as well as the deacetylation of specific lysine (Lys) residues by Sirt1, which is critical for TTF-1’s role in regulating orexin 2 receptor expression in the hypothalamus [28], we further aimed to assess the impact of energy deficiency on the physical interaction between Sirt1 and TTF-1. Additionally, we examined the effect of this interaction on TTF-1 acetylation levels, as depicted in Figure 2A,D. Fasting for 24 h increased the interaction between TTF-1 and Sirt1 (Figure 2B) and resulted in a decreased level of acetylated TTF-1 (Figure 2C) in the mouse hypothalamus, suggesting that energy deficiency induces an increase in the deacetylated TTF-1 level. The in vitro treatment of cultured mHypoA cells with a starvation medium yielded comparable findings regarding the interaction between Sirt-1 and TTF-1, as well as the levels of acetylated TTF-1 (Figure 2D–F). These results suggest that energy deficiency activates the interaction between Sirt1 and TTF-1 and thus stimulates TTF-1 deacetylation.

### 2.3. Modulation of Sirt1–TTF-1 Interaction and TTF-1 Deacetylation by a Sirt1 Activator and Inhibitor

We investigated the effects of Sirt1 activation on the interaction between Sirt1 and TTF-1, as well as the deacetylation of TTF-1, given the crucial role of Sirt1 in modulating the acetylation state of several different transcription factors [30,31]. To activate Sirt1, we utilized resveratrol (RSV), a well-known Sirt1 activator that increases NAD^+^ levels, essential for Sirt1 activation [32,33,34] (Figure 3A–F). Treatment with RSV resulted in an increase in the NAD^+^/NADH ratio (Figure 3A), along with the stimulation of TTF-1 and Sirt1 expression (Figure 3 C,D). Furthermore, the RSV treatment also promoted the interaction between TTF-1 and Sirt1 in the cultured mHypoA cells (Figure 3E). However, the RSV treatment resulted in a decrease in the level of acetylated TTF-1 (Figure 3F), suggesting a link between Sirt1 activation and TTF-1 deacetylation. 

To further confirm the role of Sirt1 in TTF-1 deacetylation, we next determined the effect of the Sirt1 inhibitor EX527 on the energy-deficiency-induced TTF-1 deacetylation (Figure 3G–K). The expression levels of both TTF-1 and Sirt1 were upregulated in response to the starvation medium, but their increases were significantly inhibited by treatment with EX527 (Figure 3G–I). The starvation-medium-induced increase in interaction between TTF-1 and Sirt1 was also decreased by the EX527 treatment (Figure 3J). Moreover, the treatment with EX527 diminished the effect of the starvation medium on the decrease in acetylated TTF-1 level (Figure 3K). Therefore, these results together suggest that activation of Sirt1 is important for the energy-deficiency-induced deacetylation of TTF-1.

### 2.4. Modulation of TTF-1 Nuclear Translocation by a Sirt1 Activator and Inhibitor

Previous studies have provided evidence that alterations in TTF-1 acetylation have an impact on both the DNA binding affinity and transcriptional activity of TTF-1 [35,36,37], which occur following the nuclear translocation of TTF-1. To further confirm this effect, we conducted immunocytochemistry (Figure 4A) and Western blot analysis (Figure 4B,E) on mHypoA cells after treating them with either a Sirt1 activator, RSV, or a Sirt1 inhibitor, EX527.

Treatment with RSV resulted in an enhanced nuclear translocation of TTF-1 (Figure 4C), while no significant effect on the nuclear translocation of Sirt1 was observed (Figure 4D). Conversely, the administration of EX527 strongly inhibited the increase in nuclear translocation of both TTF-1 and Sirt1 induced by the starvation medium (Figure 4E–G). These findings strongly indicate that Sirt1-induced deacetylation regulates the nuclear translocation of TTF-1 under conditions of energy deficiency.

### 2.5. Importance of Acetyl-Lys in the Regulation of Target Gene Expression by TTF-1

Lys182 is a critical site for regulating acetylation and deacetylation within the homeodomain of TTF-1. Its acetylation state has a significant impact on the transcription of target genes [36,37]. Specifically, Sirt1-mediated deacetylation targets acetyl-Lys182, activating the functional role of TTF-1 [28]. To assess the importance of Lys182 acetylation in TTF-1’s activity with respect to target gene expression, we generated a mutant form of TTF-1 known as the K182Q mutant, in which the Lys182 residue was replaced with glutamine (Gln). This substitution of Gln neutralizes the positive charge of Lys, mimics the structural characteristics of acetyl-Lys, and counteracts the deacetylated TTF-1 [28]. In accordance with this, a TTF-1-induced increase in AgRP promoter activity was attenuated in the K182Q mutant (Figure 5A). Moreover, the TTF-1-induced decrease in POMC promoter activity was inhibited by the K182Q mutant (Figure 5B). By contrast, other mutants did not affect the role of TTF-1 in AgRP and POMC transcription (Appendix A). To further validate the significance of Lys182 for the impact of TTF-1 on target gene expression, we assessed the promoter activities of AgRP and POMC regulated by the K182Q mutant of TTF-1 under the influence of a Sirt1 activator (RSV) or inhibitor (EX527) in the mHypoA cells. The activity of the AgRP promoter was enhanced by RSV and reduced by EX527 (Appendix A), whereas the activity of the POMC promoter was diminished by RSV and augmented by EX527 (Appendix A). RSV promoted TTF-1′s effect on increasing AgRP and reducing POMC luciferase activity (Figure 5C,D), whereas EX527 hindered these TTF-1 activities (Figure 5E,F). However, the K182Q mutant of TTF-1 displayed reduced responses to the stimulatory effect of RSV on AgRP promoter activity and the inhibitory effect of RSV on POMC promoter activity (Figure 5C,D). Conversely, the K182Q mutant exhibited enhanced responses to the effects of EX527 on the target gene expressions (Figure 5E,F). Other TTF-1 mutants (K161Q, K179Q, and K215Q) did not affect the RSV- and EX527-induced role of TTF-1 in the promoter activities of the AgRP (Appendix A) and POMC (Appendix A) genes. Collectively, these findings highlight the significant role of Sirt1-dependent deacetylation at Lys182 in the functional activity of TTF-1, specifically in the regulation of AgRP and POMC gene expression. 

### 2.6. Impact of Lys182 Mutation on TTF-1 Activities in Regulating Target Gene Expression

To further confirm the significance of Lys182 on the impact of TTF-1 on target gene expression, we introduced another mutation in TTF-1, substituting Lys182 with arginine (K182R). Previous studies indicated that Lys-to-arginine mutants maintain a positive charge and are incapable of being acetylated, effectively mimicking a deacetylated state of Lys [38,39]. In accordance with this, the K182R mutant of TTF-1 enhanced AgRP promoter activity (Figure 6A) and AgRP mRNA expression (Figure 6C) and diminished POMC promoter activity (Figure 6B) and POMC mRNA expression (Figure 6D), which are opposite effects to those of the K182Q mutant of TTF-1, which mimics acetylated TTF-1. 

In this study, we found that the acetylation state of TTF-1 affects its nuclear translocation. Therefore, we next determined the effect of Lys182 mutants on the nuclear translocation of TTF-1 using immunocytochemistry (Figure 6E) and Western blotting (Figure 6F). The nuclear translocation of TTF-1 was increased by the K182R mutant and decreased by the K182Q mutant (Figure 6H,I). These results indicated that the K182R mutant of TTF-1 replicates the nuclear translocation mechanism and target gene expression regulation observed in RSV-induced TTF-1. By contrast, the K182Q mutant exhibits similarities to the activity of EX527-induced TTF-1. Therefore, these findings suggest that Sirt1-induced deacetylation at Lys182 is crucial for the functional activity of TTF-1. 

## 3. Discussion

TTF-1 has been widely implicated in the hypothalamic regulation of energy homeostasis [4,6,8,9,40]. TTF-1 acts as an appetite-driving transcription factor that is stimulated in response to a decrease in body energy levels. However, the specific mechanisms by which TTF-1 responds to energy deficiency have not been sufficiently elucidated. In this study, we aimed to unmask the role of NAD^+^–Sirt1-induced deacetylation of TTF-1 in its response to energy deficiency.

Energy deficiency increases the NAD^+^/NADH ratio and activates Sirt1 expression, indicating the involvement of the cellular redox balance and NAD^+^ metabolism in response to low energy conditions [13,41,42]. Consistent with these observations, we found that energy deficiency increased the NAD^+^/NADH ratio, accompanied by upregulated Sirt1 expression. Importantly, this energy-deficiency-induced response resulted in the deacetylation of TTF-1. To further investigate the relationship between NAD^+^–Sirt1 signaling and TTF-1 response, we employed RSV and EX527 as pharmacological tools. RSV, an activator of Sirt1, increases intracellular NAD^+^ levels through adenosine monophosphate-activated protein kinase (AMPK) [32,33,34]. However, EX527 selectively inhibits Sirt1 by obstructing its NAD^+^ binding site [43,44]. Our results demonstrated that RSV treatment enhanced the NAD^+^/NADH ratio, stimulated Sirt1 and TTF-1 expression, and facilitated the interaction between Sirt1 and TTF-1. Additionally, RSV-induced activation of Sirt1 promoted the deacetylation and nuclear translocation of TTF-1, leading to enhanced TTF-1 activity in regulating target gene expression. Conversely, EX527-mediated Sirt1 inhibition disrupted the energy-deficiency-induced interaction between Sirt1 and TTF-1, impeded TTF-1 deacetylation, and consequently hindered the regulation of its target genes, including AgRP and POMC.

The acetylation and deacetylation of transcription factors are critical post-translational modifications that can modulate their activity, including DNA binding affinity and nuclear translocation [45,46,47,48]. Acetylation can either enhance or diminish transcription factor activity, depending on the context [35,49], while deacetylation often stimulates transcription factor activity [28,36,50,51]. TTF-1 acetylation has been shown to modulate its activity in the regulation of several genes in different tissues [35,36,37,52,53]. A previous study has observed that deacetylated TTF-1 induced by Sirt1 upregulated the expression of the hypothalamic orexin 2 receptor, which plays a vital role in regulating emotions, feeding, metabolism, respiration, and sleepiness [28]. Furthermore, we have demonstrated that the nuclear translocation and functional activity of TTF-1 in regulating target gene expression are inhibited by EX527-mediated Sirt1 inhibition and a TTF-1 mutation (K182Q) that negates deacetylation. Thus, NAD^+^–Sirt1-induced deacetylation of TTF-1 emerges as a critical factor in its activity for nuclear translocation and the regulation of AgRP and POMC gene expression. 

Interestingly, the acetylation state of TTF-1 can exert opposite effects on different target genes in different tissues, suggesting the presence of tissue-specific factors and context-dependent regulation [35,36,37,52,53]. Further investigations are warranted to unmask the precise mechanisms by which the acetylation state of TTF-1 exerts distinct effects on target gene regulation in a tissue-specific manner. TTF-1 contains conserved acetylation and phosphorylation sites within its homeodomain, suggesting that it may interact with additional co-factors that regulate its post-translational modifications [36,37]. Future investigations should aim to identify these co-factors and delineate their influence on TTF-1’s post-translational modifications, including phosphorylation, sumoylation, and acetylation, in response to energy status.

In conclusion, our study provides novel insights into the dynamic regulation of TTF-1 through NAD^+^–Sirt1-induced deacetylation in response to energy deficiency. This process plays a critical role in maintaining the functional activity of TTF-1, specifically in the regulation of AgRP and POMC gene expression. These findings broaden our understanding of the mechanisms involved in energy sensing and the ability to respond to fluctuations in energy levels within the hypothalamus. Future studies that identify other co-factors or modifications that may influence TTF-1 activity may provide a more comprehensive understanding of the regulatory mechanisms at play. Furthermore, the investigation of how TTF-1 is dysregulated in conditions such as obesity, diabetes, or eating disorders may uncover potential therapeutic targets for modulating energy balance and metabolic health.

## 4. Materials and Methods

### 4.1. Animals

Eight-week-old male C57BL/6 mice were purchased from Koatech (Namyangju-si, Republic of Korea) and used for animal experiments. Animals were fed a standard diet (Koatech) ad libitum and given free access to tap water. All animals were maintained in temperature- and humidity-controlled rooms with a 12 h/12 h light-dark cycle, with the lights on from 7:00 a.m. to 7:00 p.m. and a regulated temperature (23–25 °C). All animals and procedures used were in accordance with the guidelines and approval of the Institutional Animal Care and Use Committee of the University of Ulsan (BJL-20-030). For the fasted study, food was removed for 24 h starting at 10:00 a.m. with water available ad libitum to create a fasting condition for the animals (n = 4~6), while control animals (n = 4~6) had free access to food and water during the experiment.

### 4.2. Cell Culture and Transfection

To verify the mechanisms by which TTF-1 responds to changes in energy states in hypothalamic neurons, mHypoA cells (CELLutions Biosystems Inc., Burlington, ON, Canada) that were originally generated from the hypothalamus and contained a broad library of hypothalamic neuronal phenotypes were used [54,55]. mHypoA cells were maintained in high-glucose Dulbecco’s modified Eagle’s medium (Welgene, Gyeongsan-si, Republic of Korea) supplemented with 10% fetal bovine serum (Welgene) and 100 U/mL penicillin–streptomycin (Welgene) in a humidified atmosphere with 5% CO_2_ at 37 °C. To determine the effects of Sirt1 on TTF-1 expression and acetylation, mHypoA cells were treated for 24 h with either RSV (100 μM, Enzo Life Sciences, Inc., Farmingdale, NY, USA) or EX527 (100 μM, Sellekchem, Houston, TX, USA). To investigate the effect of TTF-1, mHypoA cells were grown to 70% confluence, and they were transiently transfected with expression vectors using jetPRIME**^®^** reagent (Polyplus, New York, NY, USA), expression vector pcDNA4-V5/HIS containing the TTF-1 coding region (300 ng), or control pcDNA4-V5/HIS (Invitrogen, Gaithersburg, MD, USA). 

### 4.3. Plasmids and Site-Directed Mutagenesis

TTF-1 mutants were made sequentially by using the EZchange™ site-directed mutagenesis kit (Enzynomics, Daejeon, Republic of Korea) according to the manufacturer’s protocol and with the following primers: TTF-1 K161Q sense primer, 5′-AGC GTC GGG TGC TCT TCT-3′; TTF-1 K161Q antisense primer, 5′-GCC GGC GGG GTG CGC TGG G-3′; TTF-1 K179Q sense primer, 5′-AGC AGC AGA AGT ACC TGT-3′; TTF-1 K179Q antisense primer, 5′-GGA AGC GTC GCT CGA GCT C-3′; TTF-1 K182Q sense primer, 5′-AGT ACC TGT CGG CGC CGG-3′; TTF-1 K182Q antisense primer, 5′-GCT GCT GCT TGA AGC GTC G-3′; TTF-1 K215Q sense primer, 5′-AGC GCC AGG CGA AGG ACA-3′; TTF-1 K215Q antisense primer, 5′-GCA TCT TGT AGC GGT GGT T-3′; TTF-1 K182R sense primer, 5′-GTA CCT GTC GGC GCC GGA-3′; and TTF-1 K182R antisense primer, 5′-CTC TGC TGC TTG AAG CGT C-3′. All mutations were verified by DNA sequencing.

### 4.4. RNA Isolation and Quantitative Real-Time Polymerase Chain Reaction (qPCR)

RNA was extracted using RNAiso reagent (Takara Bio Inc., Shiga, Japan) according to the manufacturer’s protocol. RNA purity and concentration were measured using a Take3 micro-volume plate (Agilent Technologies, Inc., Santa Clara, CA, USA) and cDNA was synthesized by using MMLV reverse transcriptase (Beams Biotechnology, Gyeonggi-do, Republic of Korea) according to the manufacturer’s protocol. Real-time PCR was performed using the following primer sets: TTF-1 sense primer, 5′-AAC AGA AGT ACC TGT CGG CG-3′; TTF-1 antisense primer, 5′-ACC AGA TCT TGA CCT GCG TG-3′; Sirt1 sense primer, 5′-GAC AGA ACG TCA CAC GCC AG-3′; Sirt1 antisense primer, 5′-TTG TTC GAG GAT CGG TGC CAA-3′; AgRP sense primer, 5′-TGT GTA AGG CTG CAC GAG TC-3′; AgRP antisense primer, 5′-GGC AGT AGC AAA AGG CAT TG-3′; POMC sense primer, 5′-GAG TTC AAG AGG GAG CTG GA-3′; POMC antisense primer, 5′-GGT CAT GAA GCC ACC GTA AC- 3′; ꞵ-actin sense primer, 5′-GGG GTG TTG AAG GTC TCA AA-3′; and β-actin antisense primer, 5′-GAT CTG GCA CCA CAC CTT CT-3′. Real-time PCR experiments were performed with a BlasTaq 2X qPCR MasterMix (Applied Biological Materials Inc., Richmond, BC, Canada) using the StepOnePlus Real-Time PCR System (Thermo Fisher Scientific Inc., Waltham, MA, USA) for ~40 cycles.

### 4.5. Western Blot Analysis and Immunoprecipitation 

mHypoA cells and medial basal hypothalamus (MBH) tissues were harvested and lysed with RIPA buffer (Lugen Sci Inc., Gyeonggi-do, Republic of Korea). Protein concentration was measured with a Bradford dye-binding assay (Bio-Rad Laboratories, Inc., Hercules, CA, USA), and 30 µg of protein from each sample was separated with sodium dodecyl sulfate–polyacrylamide gel electrophoresis and transferred onto nitrocellulose membranes (GE Healthcare Life Science, Madison, WI, USA) by electrophoretic transfer. The membrane was blocked with 5% bovine serum albumin in Tris-buffered saline–tween buffer and then incubated with rabbit anti-TTF-1 antibody (#ab76013, Abcam, Inc., Cambridge, UK; 1:1000 dilution), rabbit anti-Sirt1 antibody (Sigma-Aldrich, St. Louis, MO, USA; 1:2000 dilution), rabbit anti-acetyl-Lys antibody (#9441, Cell Signaling Technology, Danvers, MA, USA; 1:1000 dilution), and goat anti-lamin B antibody (#sc-6216, Santa Cruz Biotechnology, Dallas, TX, USA; 1:1000 dilution). The membrane was incubated with horseradish peroxidase (HRP)-conjugated mouse secondary antibody (#sc-516102, Santa Cruz Biotechnology; 1:4000 dilution), HRP-conjugated rabbit secondary antibody (#sc-2357, Santa Cruz Biotechnology; 1:4000 dilution), and HRP-conjugated goat secondary antibody (#sc-2354, Santa Cruz Biotechnology; 1:4000 dilution). The immunoreactive signals were detected with a chemiluminescent detection reagent (Thermo Fisher Scientific Inc.). Protein density was normalized using an anti-ꞵ-actin antibody (#A5441, Sigma-Aldrich; 1:5000 dilution). 

To analyze the effect of changes in the energy state on the TTF-1 and Sirt1 interaction and TTF-1 acetylation, we performed immunoprecipitation. A 500-µg sample of protein extract was immunoprecipitated overnight at 4 °C with mouse anti-TTF-1 antibody (#MA5-13961, Invitrogen) and rabbit anti-Sirt1 antibody (#07-131, Sigma-Aldrich), or mouse IgG (#sc-516102, Santa Cruz Biotechnology) and rabbit IgG (#sc-2357, Santa Cruz Biotechnology) as negative controls. This was followed by incubation with Dynabeads protein G (Invitrogen) for immunoprecipitation for 48 h at 4 °C. Immunoprecipitates were washed 5 times each with 650 µL of RIPA buffer with protease inhibitor (Roche, Basel, Switzerland) using a rotator. The protein band intensities were measured using ImageJ software (National Institutes of Health, Bethesda, MD, USA, v1.54d), the band intensity of Sirt1 protein was standardized against the TTF-1 band, and the band intensity of TTF-1 protein was standardized against the Sirt1 band. Based on this, a comparison of band intensities was made between the control group and the experimental group. To analyze TTF-1 acetylation, the band of acetyl-Lys protein was standardized based on the TTF-1 band for quantification, and the band strengths of the control group and the experimental group were compared. 

### 4.6. NAD^+^ and NADH Measurements

To determine the effects of changes in energy state, mice were either fasted for 24 h or fed ad libitum before being sacrificed. The MBH from the brain was dissected and collected according to the brain atlas under a stereomicroscope. mHypoA cells were treated with 5 mM glucose without serum for 24 h, and then the cells were scraped. NAD+ and NADH levels were quantified by using the EnzyFluo™ NAD+:NADH assay kit (BioAssay Systems, Hayward, CA, USA) according to the manufacturer’s protocol.

### 4.7. Immunocytochemistry (ICC)

mHypoA cells were fixed in 4% paraformaldehyde for 20 min, washed with phosphate-buffered saline (PBS, Sigma-Aldrich), and incubated for 30 min in 0.3% Triton-X 100. After several washes with PBS, the cells were incubated with mouse anti-TTF-1 antibody (#MA5-13961, Invitrogen; 1:1000 dilution) and rabbit anti-Sirt1 antibody (#07-131, Sigma-Aldrich; 1:1000 dilution) overnight at 4 °C. After several PBS washes, the cells were incubated with anti-rabbit Alexa Fluor 488 (Thermo Fisher Scientific Inc.; 1:2000 dilution) and anti-mouse Alexa Fluor 594 (Thermo Fisher Scientific Inc.; 1:2000 dilution) secondary antibodies on a shaker for 2 h at room temperature. Subsequently, the coverslips were treated with DAPI (#P36962; Thermo Fisher Scientific Inc.) and the fluorescence expression was captured with an FV1200 confocal laser scanning microscope (Olympus America, Inc., Center Valley, PA, USA). 

### 4.8. Promoter Assays

To determine whether Sirt1 and TTF-1 regulate AgRP and POMC transcription, mHypoA cells were co-transfected with an AgRP promoter-luciferase reporter vector (500 ng) or a POMC promoter-luciferase reporter vector (500 ng) and a TTF-1 expression vector (300 ng) using jetPRIME**^®^** reagent (Polyplus). Transfection efficiency was normalized by co-transfecting a Renilla reporter plasmid (pRL-SV40 vector; Promega, Madison, WI, USA) at 100 ng/well. Transfected cells were harvested 24 h after transfection and luciferase activity was measured using a luciferase reporter assay system (Promega) according to the manufacturer’s protocols.

### 4.9. Statistical Analyses

Statistical analyses were performed with GraphPad Prism 9 software (GraphPad Software, San Diego, CA, USA). All data are expressed as the mean ± standard error of measure. Data were analyzed with a one-way or two-way analysis of variance followed by Tukey’s multiple comparison tests for unequal replications. A Student’s *t*-test was used to compare the two groups. All data were confirmed to meet the normality assumption using the Shapiro–Wilk test.

## 5. Conclusions

In this study, we elucidated the mechanism by which energy deficiency activates Sirt1, promoting the deacetylation of TTF-1, and subsequently, regulating the transcription of AgRP and POMC to control energy metabolism (Figure 7). In situations of energy deficiency, NAD^+^-dependent Sirt1 deacetylates TTF-1, leading to the transcriptional regulation of AgRP and POMC, thus maintaining energy homeostasis (Figure 7). 

## Figures and Tables

**Figure 1 ijms-24-12530-f001:**
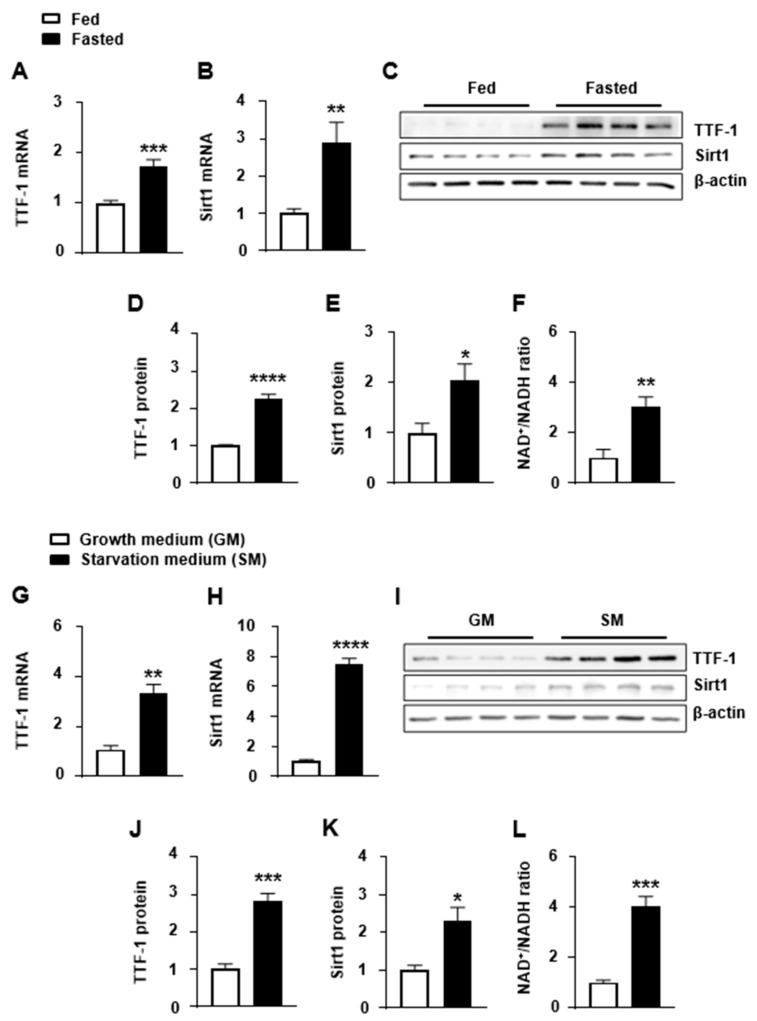
Energy-deficiency-induced change in Sirt1 and TTF-1 expression. (**A**–**F**) Mice were fasted for 24 h (Fasted) and compared to mice with ad libitum access to food (Fed). (**G**–**L**) mHypoA cells were treated with growth medium (GM) containing 25 mM glucose and 10% fetal bovine serum or serum-free starvation medium (SM) containing 5 mM glucose for 24 h. TTF-1 (**A**,**G**) and Sirt1 (**B**,**H**) mRNA levels were analyzed with real-time qPCR and normalized to ꞵ-actin mRNA (n = 4–6/group). Protein was extracted from MBH tissues (**C**–**E**) and mHypoA cells (**I**–**K**) and was measured with Western blot analyses (**C**,**I**). Western blot bands for TTF-1 (**D**,**J**) and Sirt1 (**E**,**K**) protein expression were calculated after normalization with the β-actin level (n = 4/group). Intracellular NAD^+^ and NADH levels were measured using a microplate reader for MBH tissues (**F**) (n = 5/group) and mHypoA cells (**L**) (n = 4/group) and the NAD^+^/NADH ratio was calculated. * *p* < 0.05, ** *p* < 0.01, *** *p* < 0.001, **** *p* < 0.0001. All data represent arbitrary units.

**Figure 2 ijms-24-12530-f002:**
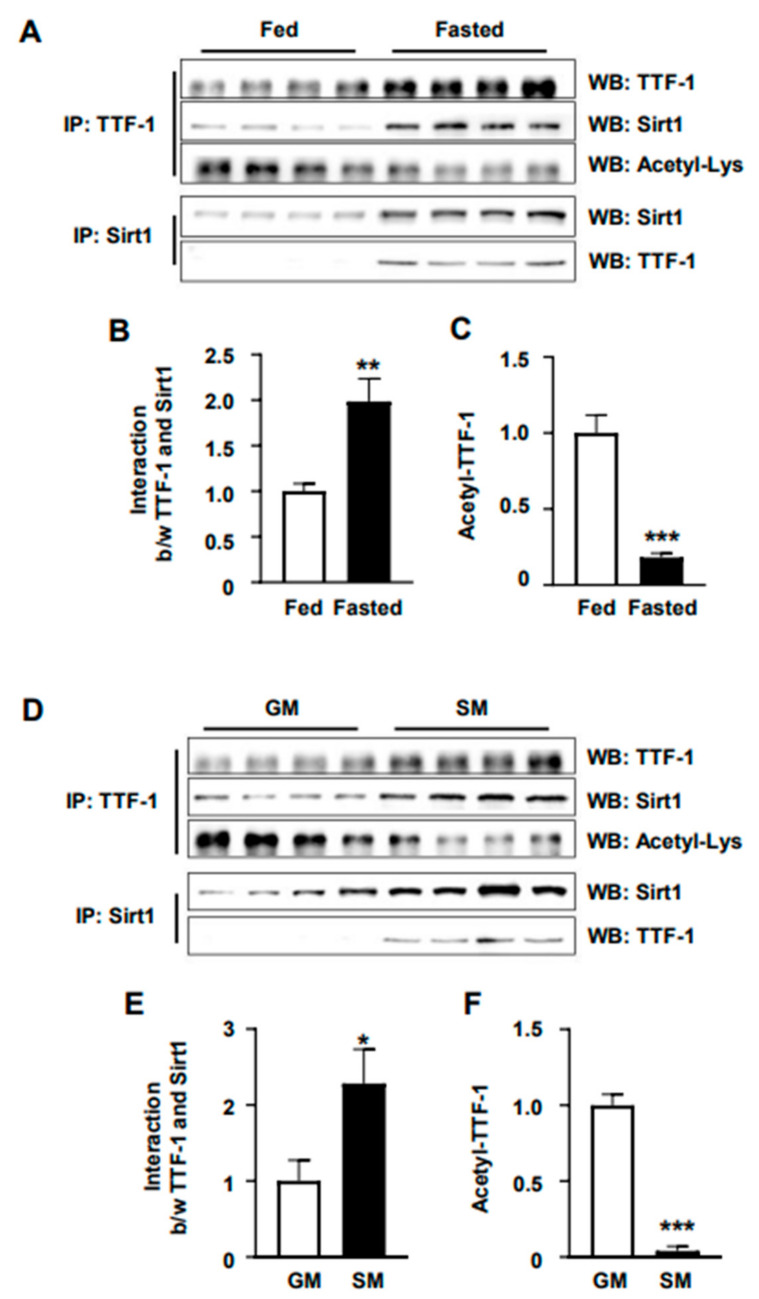
Change in TTF-1 acetylation by energy state. To determine the interaction between Sirt1 and TTF-1 and the acetylation of TTF-1 in different energy states, levels of TTF-1 protein and acetyl-Lys were investigated by Western blot (WB) analysis using protein samples. Protein was extracted by immunoprecipitation (IP) with TTF-1 antibody (IP: TTF-1) or Sirt1 antibody (IP: Sirt1) from the hypothalami (**A**–**C**) of mice that were fasted for 24 h (Fasted) or fed with ad libitum access to food (Fed) and from mHypoA cells (**D**–**F**) that were treated with growth medium (GM) or starvation medium (SM). Levels of interaction between (b/w) TTF-1 and Sirt1 (**B**,**E**) and acetylated TTF-1 (Acetyl-TTF-1) (**C**,**F**) were normalized with the amount of IP TTF-1 or IP Sirt1 for each construct relative to the control conditions (Fed or GM) (n = 4/group). To analyze the interaction between TTF-1 and Sirt1, the samples were immunoprecipitated using their respective antibodies. Subsequently, a positive control antibody and an antibody specific to the interacting protein were used for detection. For quantification, the band intensity was normalized using the band of a positive control, and then the band intensities of the experimental group were compared with those of the control group. * *p* < 0.05, ** *p* < 0.01, *** *p* < 0.001. All data represent arbitrary units.

**Figure 3 ijms-24-12530-f003:**
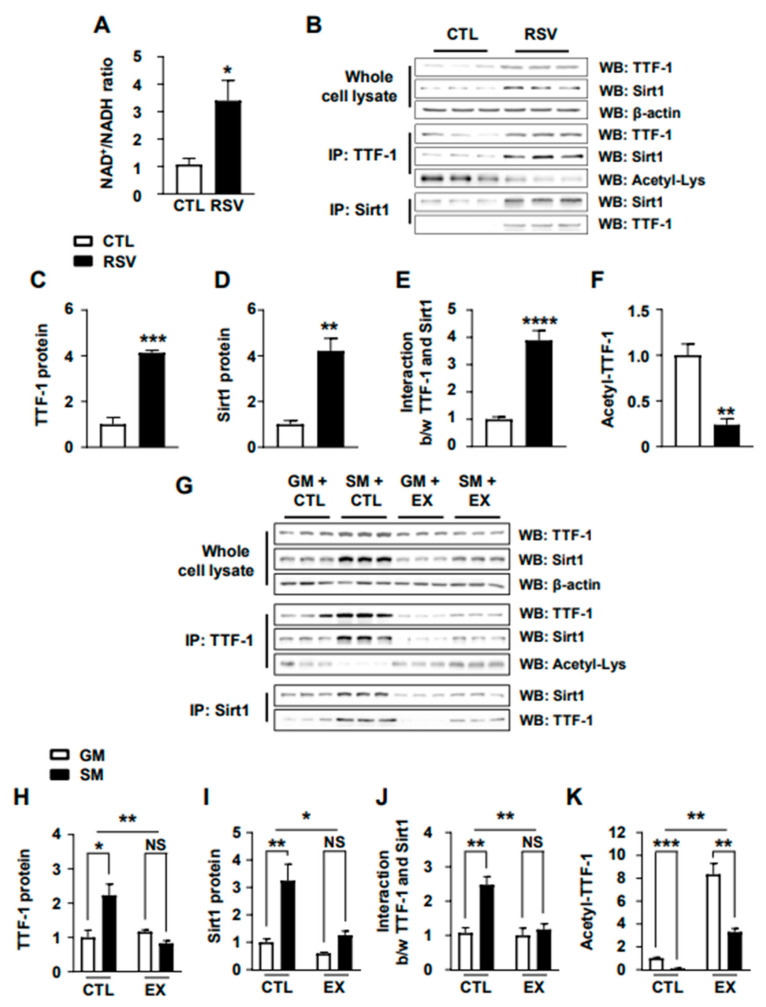
Effect of resveratrol and EX527 on Sirt1–TTF-1 interaction and TTF-1 deacetylation. mHypoA cells were treated with resveratrol (RSV) or EX527 (EX) and their levels of TTF-1, Sirt1, and acetyl-Lys were determined with Western blotting (WB) after immunoprecipitation (IP) using TTF-1 (IP: TTF-1) and Sirt1 (IP: Sirt1) antibodies. (**A**) The effect of RSV on cells was verified by the determination of intracellular NAD^+^ and NADH levels (n = 3/group). (**B**–**F**) mHypoA cells were cultured in GM for 24 h and were treated with RSV for 24 h, and their TTF-1, Sirt1, and acetyl-Lys levels were determined with WB. (**G**–**K**) mHypoA cells cultured in SM for 24 h were treated with EX for 24 h and their TTF-1, Sirt1, and acetyl-Lys levels were measured with WB. TTF-1 (**C**,**H**) and Sirt1 (**D**,**I**) protein levels were calculated from WB bands after normalization with the β-actin level (n = 3). The level of interaction between TTF-1 and Sirt1 (**E**,**J**) and the acetyl-Lys (**F**,**K**) level were normalized to the amount of IP TTF-1 and/or IP Sirt1 for each construct relative to the normal conditions (GM + CTL) (n = 3). * *p* < 0.05, ** *p* < 0.01, *** *p* < 0.001, **** *p* < 0.0001. NS = not significant. All data represent arbitrary units.

**Figure 4 ijms-24-12530-f004:**
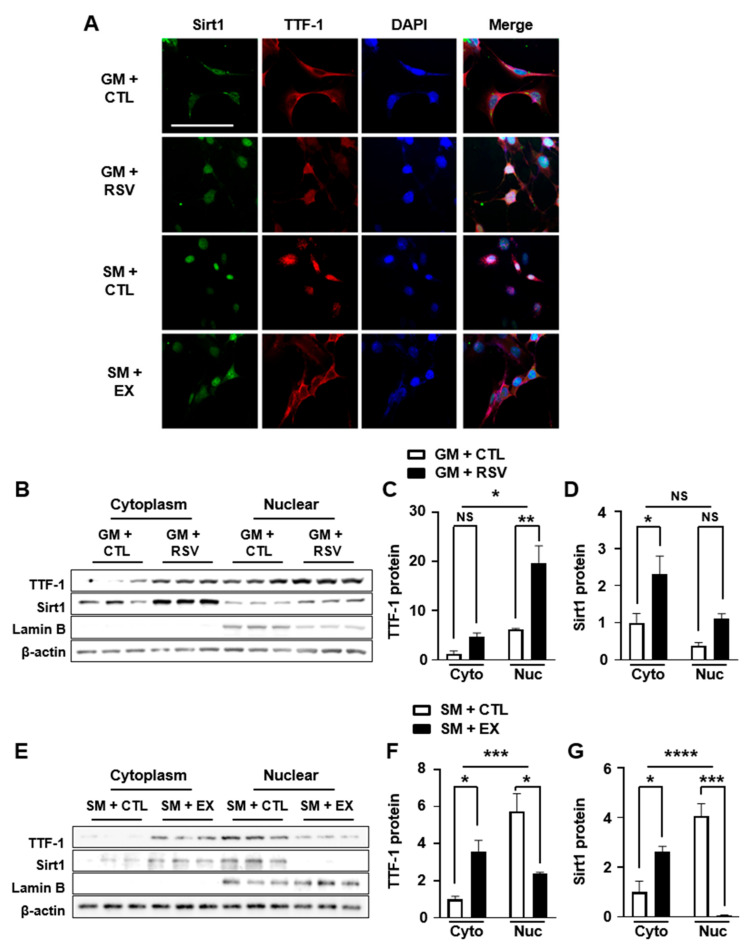
Effect of a Sirt1 activator and a Sirt1 inhibitor on TTF-1 nuclear translocation. To determine the effect of Sirt1-induced deacetylation on the nuclear translocation of TTF-1, we performed immunocytochemistry (**A**) and Western blot analyses (**B**,**E**) using TTF-1 and Sirt1 antibodies on mHypoA cells. (**B**–**G**) The mHypoA cells were cultured in GM or SM and treated with a Sirt1 activator (RSV) for 24 h (**B**–**D**) or a Sirt1 inhibitor (EX) for 24 h (**E**–**G**). Western blot analyses were conducted using cytoplasmic and nuclear extracts from cultured mHypoA cells and levels of proteins were calculated after normalization with levels of internal controls of β-actin and nuclear lamin B (n = 3/group). * *p* < 0.05, ** *p* < 0.01, *** *p* < 0.001, **** *p* < 0.0001. NS = not significant. All data represent arbitrary units. Scale bar = 100 µm.

**Figure 5 ijms-24-12530-f005:**
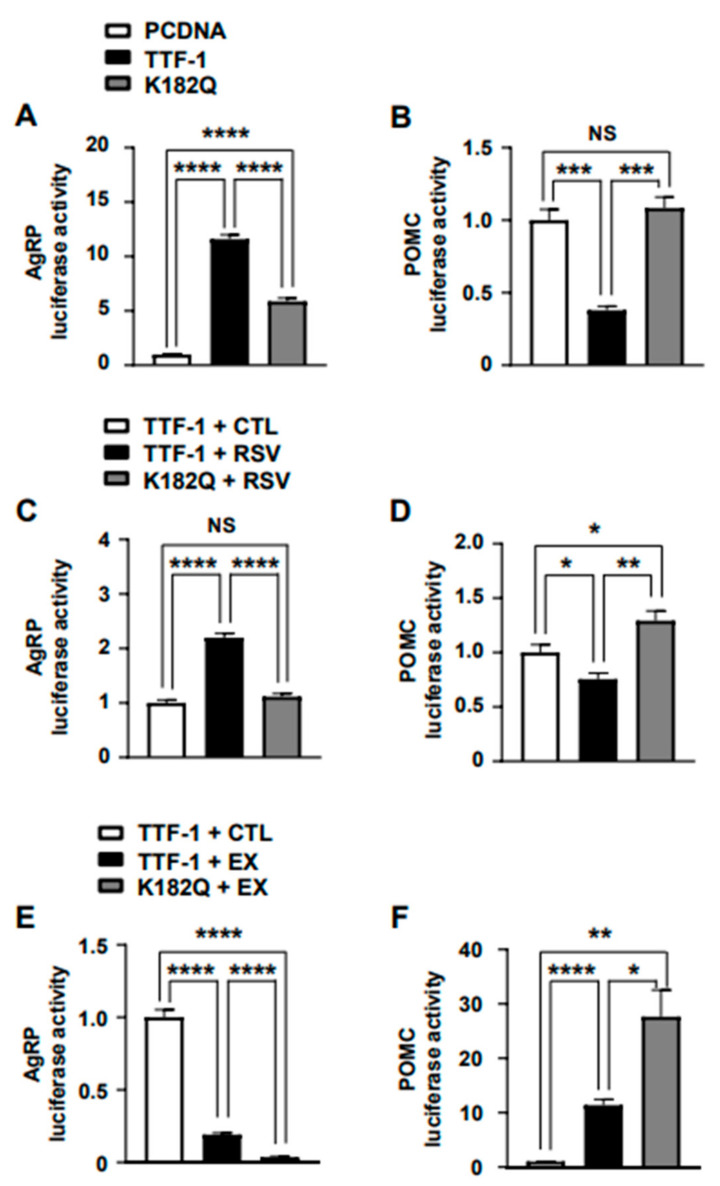
Modulation of AgRP and POMC promoter activity by mutation of TTF-1 at Lys182. To validate the importance of the deacetylation at Lys182 in TTF-1′s activity, a mutant TTF-1 with Gln replacing Lys182 (K182Q) was generated. mHypoA cells were transfected with AgRP and POMC promoter-luciferase reporter vectors and expression vectors for the K182Q mutant, the wild-type TTF-1 (TTF-1), and control pcDNA. The resulting AgRP (**A**) and POMC (**B**) promoter activities were determined with luciferase assays (n = 4/group). To verify the effect of Sirt1 activation on the role of the K182Q mutant in the AgRP and POMC promoter activities, mHypoA cells were transfected with vectors for AgRP and POMC promoter-luciferase and the K182Q mutant and treated for 24 h with RSV or EX. The resulting luciferase activities for AgRP (**C**,**E**) and POMC (**D**,**F**) promoters were measured (n = 4/group). * *p* < 0.05, ** *p* < 0.01, *** *p* < 0.001, **** *p* < 0.0001. NS = not significant. All data represent arbitrary units.

**Figure 6 ijms-24-12530-f006:**
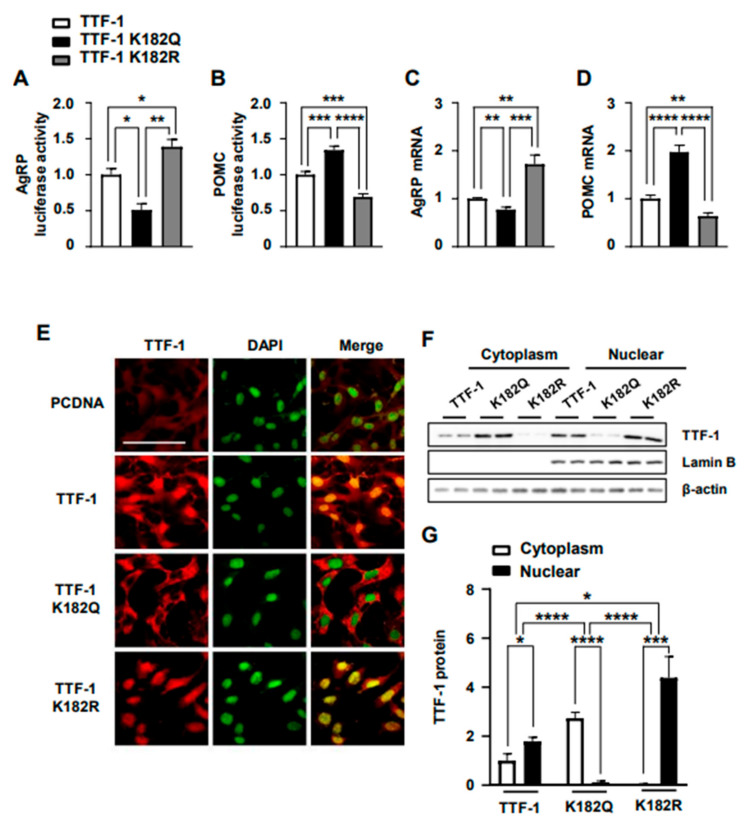
Influence of Lys182 mutation on the role of TTF-1 in the regulation of AgRP and POMC gene expression. (**A**–**D**) To identify the importance of Lys182 residue of TTF-1 in the regulation of AgRP and POMC gene expression, cells were transfected with expression vectors for wild-type TTF-1 (TTF-1) and the K182Q and K182R mutants. The AgRP (**A**) and POMC (**B**) promoter activities were determined using luciferase assays, while AgRP (**C**) and POMC (**D**) mRNA expressions were measured with real-time PCR (n = 3–6/group). (**E**) mHypoA cells were transfected with expression vectors of TTF-1, K182Q, K182R, and control pcDNA vectors, and were used for immunocytochemical observation using TTF-1 antibody. (**F**) Western blot analyses were performed to determine the amounts of cytoplasmic and nuclear TTF-1 in cells transfected with vectors for TTF-1, K182Q, and K182R. (**G**) TTF-1 protein expression was calculated after normalization with internal controls of β-actin and nuclear lamin B (n = 4/group). Scale bar = 100 μm. * *p* < 0.05, ** *p* < 0.01, *** *p* < 0.001, **** *p* < 0.0001. All data represent arbitrary units.

**Figure 7 ijms-24-12530-f007:**
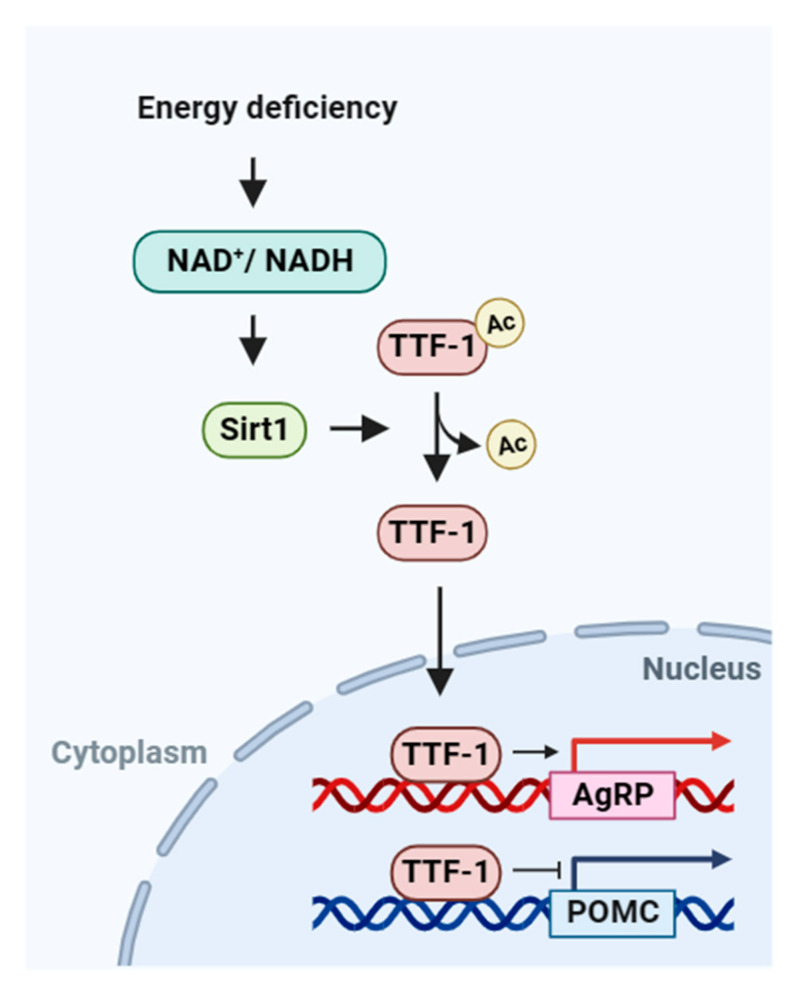
Proposed model for the Sirt1–TTF1–AgRP/POMC axis. Depending on the body’s energy status, Sirt1 regulates the deacetylation of TTF-1, which in turn modulates the expression of AgRP and POMC, thereby controlling feeding behavior.

## Data Availability

All data are reported in the manuscript.

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
