# Peer review of "Sirtuin1-Mediated Deacetylation of Hypothalamic TTF-1 Contributes to the Energy Deficiency Response"

_ijms, 2023, doi:10.3390/ijms241512530_

Round 1

Reviewer 1 Report

In the current study, authors showed evidence to prove a role for Sirt1-mediated deacethylation of TTF-1 in the hypothalamus in appetite regulation in starving mice.  

Their finding may contribute to bridging a small missing link in the starvation-induced appetite regulation, in which the involvements of Sirt1 and TTF1 as well as AgRP and POMC gene expressions have already been reported. 

Nevertheless, the impact of their finding is difficult to comprehend for non-specialist readers. Before publication in IJMS, the following concerns should be addressed.

Major concerns:

1) Please provide direct evidence to show an involvement of Sir1-mediated TTF1 deacethylation on Lys182 in the induction of AgRP and POMC gene expressions in the hypothalamus.

2) Please provide a figure that clearly illustrates the point of novelty regarding the involvement of Sirt1-TTF1-AgRP/POMC axis in appetite regulation.

Reviewer 2 Report

This study examines how TTF-1 stimulates the appetite. This study provides experimental studies with statistical analysis to indicate that  NAD+-dependent deacetylase, sirtuin1 (Sirt1), activates TTF-1 in response to energy deficiency.  This is a very well designed and very well written study that will be of interest to the field.  A few minor edits are suggested:

The results of this study are very intriguing.  To enable better search ability and citation, it is strongly recommended that the authors put in some of their quantitative results into the abstract, including significance.

I could not find where the sample sizes are listed.  The sample sizes (mice per group) should be clearly indicated in the Methods section.

The authors utilize ANOVA with t-tests corrected for multiple comparisons. This approach is very reasonable. However, it requires that the underlying data meet the normality assumption.  The authors should check the normality of their data using Shapiro Wilk or similar test to insure ANOVA can be appropriately utilized. Though unlikely, if normality is not met, the authors could easily switch to Kruskal-Wallis analysis of variance to compare medians.  The authors need to state their result of checking for normality in the methods.

Reviewer 3 Report

The manucript ijms-2542235 entitled Sirtuin1-mediated deacetylation of hypothalamic TTF-1 contributes to the energy deficiency response by Dasol Kang studies the NAD+-dependent deacetylase, sirtuin1 (Sirt1), that activates TTF-1 in response to energy deficiency.

Energy deficiency enhances the expression of both Sirt1 and TTF-1, leading to the deacetylation of TTF-1 through the interaction between the two proteins. Activation of Sirt1, induced by energy deficiency or resveratrol treatment, results in the increased deacetylation and nuclear translocation of TTF-1. Conversely, inhibition of Sirt1 prevents these Sirt1 effects. A point mutation in a lysine residue of TTF-1 disrupts its deacetylation and thus hinders its ability to regulate AgRP and POMC gene expression.

The scientific plan is clear and consistent with the available literature.

The methodology used and the experimental design is clear and well conducted.

Figures are informative and clear.

Discussion is consistent with results.

Minor comments: a linguistic revision is recommended.

Minor revision is required

Round 2

Reviewer 1 Report

In the revised manuscript, authors have fully responded to the reviewer's concerns. The current manuscript has been sufficiently improved to warrant publication in IJMS.